# IMVTS: A Detection Model for Multi-Varieties of Famous Tea Sprouts Based on Deep Learning

**Runmao Zhao** [1,2,3]🆔, **Cong Liao** [1], **Taojie Yu** [1], **Jianneng Chen** [1,2,3]🆔, **Yatao Li** [2,3,4], **Guichao Lin** [5], **Xiaolong Huan** [1,2,3] **and Zhiming Wang** [6,*]

1   School of Mechanical Engineering, Zhejiang Sci-Tech University, Hangzhou 310018, China; rmzhao@zstu.edu.cn (R.Z.); 13517960753@163.com (C.L.); 202130605347@mails.zstu.edu.cn (T.Y.); jiannengchen@zstu.edu.cn (J.C.); huanxl@zstu.edu.cn (X.H.)
2   Key Laboratory of Transplanting Equipment and Technology of Zhejiang Province, Hangzhou 310018, China; ytli@zstu.edu.cn
3   Key Laboratory of Agricultural Equipment for Hilly and Mountainous Areas in Southeastern China (Co-Construction by Ministry and Province), Ministry of Agriculture and Rural Affairs, Hangzhou 310018, China
4   School of Textile Engineering, Zhejiang Sci-Tech University, Hangzhou 310018, China
5   College of Mechanical and Electrical Engineering, Zhongkai University of Agriculture and Engineering, Guangzhou 510225, China; guichaolin@zhku.edu.cn
6   Jinhua Polytechnic Zhejiang Key Laboratory of Crop Harvesting Equipment Technology, Jinhua 321017, China
*   Correspondence: jhcwzm@163.com

**Abstract:** The recognition of fresh tea leaf sprouts is one of the difficulties in the realization of the automated picking of fresh tea leaves. At present, the research on the detection of fresh tea leaf sprouts is based on a single variety of tea leaves for a specific period or specific place, which has no advantage for the spread, promotion, and application of the methods. To address this problem, an identification of multiple varieties of tea sprouts (IMVTS) model was proposed. First, images of three different varieties of tea (ZhongCha108 (ZC108), ZhongHuangYiHao (ZH), ZiJuan (ZJ)) were obtained, and the multiple varieties of tea (MVT) dataset for training and validating models was created. In addition, the detection effects of adding a convolutional block attention module (CBAM) or efficient channel attention (ECA) module to YOLO v7 were compared. In the detection of the MVT dataset, YOLO v7+ECA and YOLO v7+CBAM showed a higher mean average precision (mAP) than YOLO v7, with 98.82% and 98.80%, respectively. Notably, the IMVTS model had the highest AP for ZC108, ZH, and ZJ compared with the two other models, with 99.87%, 96.97%, and 99.64%, respectively. Therefore, the IMVTS model was proposed on the basic framework of the ECA and YOLO v7. To further illustrate the superiority of the model, this study also conducted a comparison test between the IMVTS model and the mainstream target detection models (YOLO v3, YOLO v5, FASTER-RCNN, and SSD) and the IMVTS model on the VOC dataset, and it is clear from the test results that the mAP of the IMVTS model is ahead of the remaining models. Concisely, the detection accuracy of the IMVTS model can meet the engineering requirements for the automatic harvesting of autumn fresh famous tea leaves, which provides a basis for the future design of detection networks for other varieties of autumn tea sprouts.

**Keywords:** fresh tea leaf sprouts; multiple varieties detecting; IMVTS; YOLO v7; ECA

## 1. Introduction

China is known as the home of tea. The country's planting, output, consumption, and export volume of tea are first in the world. The extensive demand for fresh tea leaves requires effective support [1]. Famous tea is a representative of high-quality tea in China. Its raw materials are one sprout and one leaf of a tea tree. Its harvesting is still mainly manual and labor intensive, and there is a shortage of tea collectors [2]. To achieve the

intelligent picking of fresh tea leaf sprouts, researchers use machine vision technology to detect fresh tea leaves. Lei Zhang et al. [3] extracted the R, G, and B components of images of collected fresh tea leaves and then processed the adaptive threshold of the B components. A new component gray diagram was obtained in combination with the G components, and the thresholds were enhanced by segmental transformation, thereby improving the contrast between the tender leaves and the background, and then, the improved watershed function was used to obtain a good division effect. Liang Zhang et al. [4] used Bayesian inference to judge the principles and Bayes methods to build an identification model of the fresh leaf collection status in order to detect purple rose tea tree sprouts in April in real time. Although the above machine learning technology can accurately identify fresh tea leaf sprouts, the identified fresh leaf image background environment is relatively single, and the segmentation accuracy is greatly affected by the characteristics of the fresh tea leaf sprouts. It is often difficult to overcome this shortcoming.

With the rapid development of artificial intelligence technology, the use of deep-learning [5] target detection algorithms, such as Fast R-CNN [6], Faster R-CNN [7], You Only Look Once (YOLO) [8], and Single Shot MultiBox Detector (SSD) [9], etc., are increasingly applied in the agricultural field. In terms of fresh tea leaf sprout recognition, Yatao Li et al. [10] used the YOLO v3 network to detect and identify fresh tea leaf sprouts on an RGB-D image collected with an RGB-D camera and estimated the three-dimensional coordinates of the fresh tea leaf picking position in the corresponding depth chart of the enclosure. It reached 83.18%. Wenkai Xu et al. [11] proposed a detection and classification method for the two-level integration network of the variability domain. This method combined YOLO v3's fast detection capacity and the high-precision classification ability of DenseNet201 to achieve the accurate detection of fresh tea leaves. Yu-Ting Chen et al. [12] detected the OTTL area in a tea tree image through the Faster R-CNN model and then identified the picking point in the OTTL area with the FCN model to determine the three-dimensional coordinates of the picking point. The YOLO model is currently a widely used detection network. It adopts a one-stage algorithm. The network operates quickly, the memory occupation is small, and it has a fast detection speed [13]. Based on inheriting the advantages of the original YOLO model, YOLO v7 [14] has a higher detection accuracy and faster reasoning speed compared to the previous YOLO series models because of its more complex network structure and training techniques. However, because the YOLO v7 model uses only one main convolutional neural network to predict the category and location of different targets, it also improves the speed and makes the recognition accuracy lower. This may lead to more missed inspections and misunderstandings when using YOLO v7 to detect fresh tea leaf images. At the same time, the applicability of the models proposed in the existing literature to fresh sprouts of different varieties of tea is unknown.

Consequently, this study used the YOLO v7 model as the basic framework to research the identification of multiple varieties of fresh tea leaf sprouts and explored the impact of integrating CBAM and ECA mechanisms on YOLO v7 detection accuracy and generalization effects. According to the results of the research, an IMVTS model was proposed to solve the problem of high-precision detection of multiple tea species. In addition, to demonstrate the superiority of the proposed IMVTS model in detection, comparison tests of the IMVTS model with mainstream target detection models (YOLO v3, YOLO v5, FASTER-RCNN, and SSD) and detection tests of the IMVTS model on VOC datasets were designed.

## 2. Materials and Methods

### 2.1. Experimental Data Collection and Preparation

#### 2.1.1. Fresh Tea Leaf Image Collection and MVT Dataset Production

At the experimental base of the Tea Research Institute of the Chinese Academy of Agricultural Sciences (Figure 1), a total of 277 fresh leaf images of 3 tea varieties were collected, of which 90 images were ZhongCha108 (ZC108), 150 were ZhongHuangYiHao (ZH), and 37 were ZiJuan (ZJ). The sampling date was 21 September 2022. The collected fresh tea leaf images were made into a dataset of fresh leaves of multiple tea varieties that

could be used to train and verify the network model and were named the MVT dataset. An iQOO Neo3 mobile phone camera was used. The shooting angle was 30~60°, and the shooting distance was 30~50 cm. The images were stored in a JPG format, and the MVT dataset images are shown in Figure 2. The open-source labeling tool LabelImg was used to artificially label the sprout position on the fresh tea leaf images, and the labeled result was saved in the PASCAL VOC format.

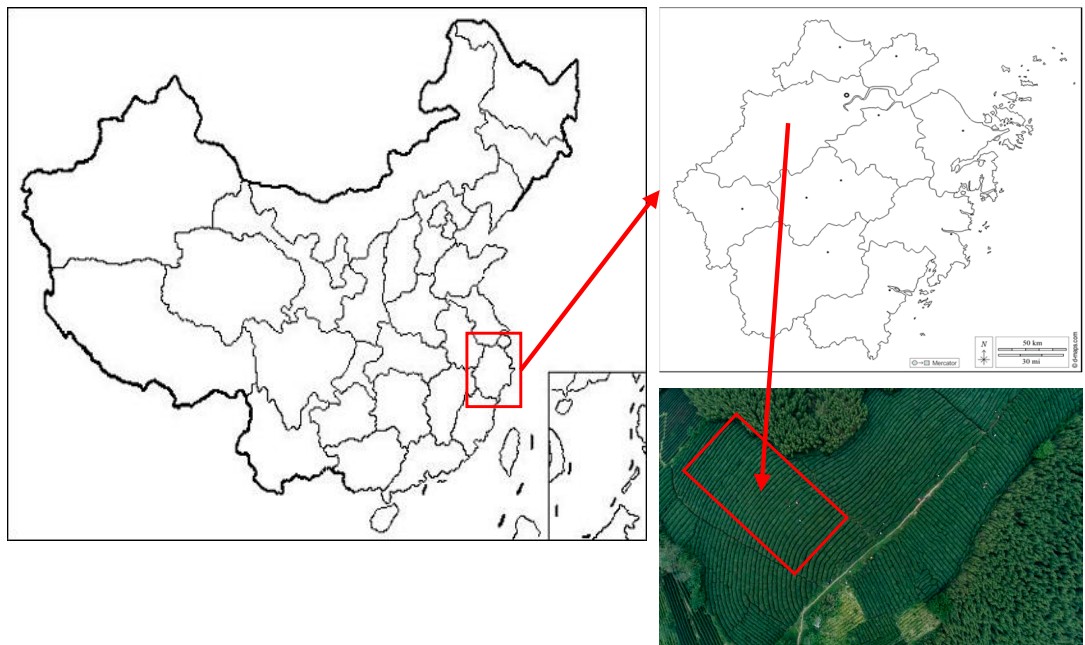

**Figure 1.** Experimental Base of the Tea Research Institute of the Chinese Academy of Agricultural Sciences.

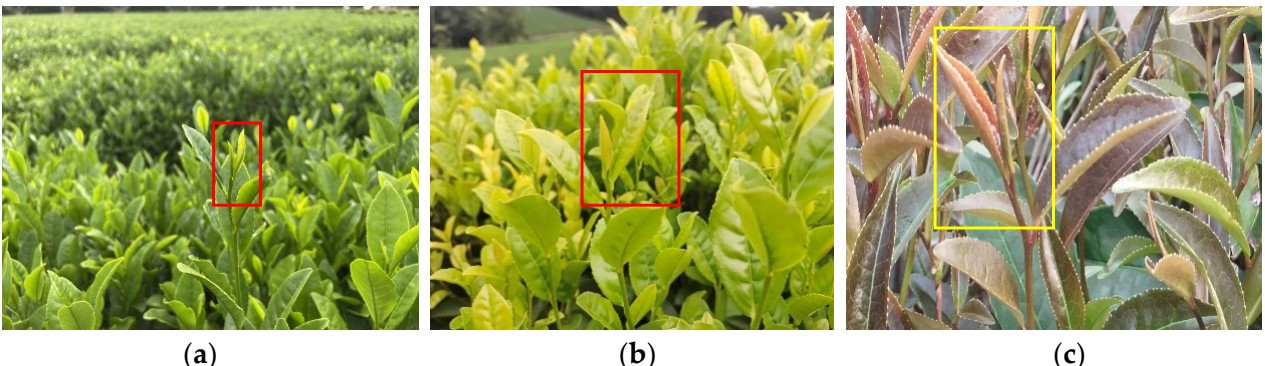

| (**a**) | (**b**) | (**c**) |

**Figure 2.** MVT dataset images; (**a**) ZC108; (**b**) ZH; (**c**) ZJ.

The images of ZC108, ZH, and ZJ as the research objects are shown in Figure 2a–c. The leaves of ZC108 are long oval in shape, green in color, slightly rumbled on the leaf surface, and flat on the leaf body, and the sprouts are yellowish-green with less velvet. The leaves of ZH are oval in shape, yellowish green in color, slightly elevated on the leaf surface, slightly inflexed on the leaf body, and the sprouts are slender. ZJ is a large-leaved, medium-sprout species. The leaf blade is upwardly slanting, willow-shaped, with an acuminate tip, and purple in color. Its petiole is purple-red in color. These images can also provide data support for future studies of other tea recognition models.

### 2.1.2. MVT Dataset Division

The 277 obtained images and corresponding labels were divided into training sets and validation sets according to the proportion of training sets: validation sets (=9:1) so that the

model training and validating in the later stage could be performed. To enrich the diversity of samples and improve the robustness of the IMVTS model, the dataset was expanded to 15,000 pieces of the original dataset, with increased or decreased brightness, increased and decreased color saturation, and horizontal flip. The application process is shown in Figure 3. After the enhancement, the number of sprouts of the fresh leaves of the three tea varieties reached 59,337, of which the number of sprouts of ZC108 was 7429, the number of sprouts of ZH was 18,902, and the number of sprouts of ZJ was 33,006.

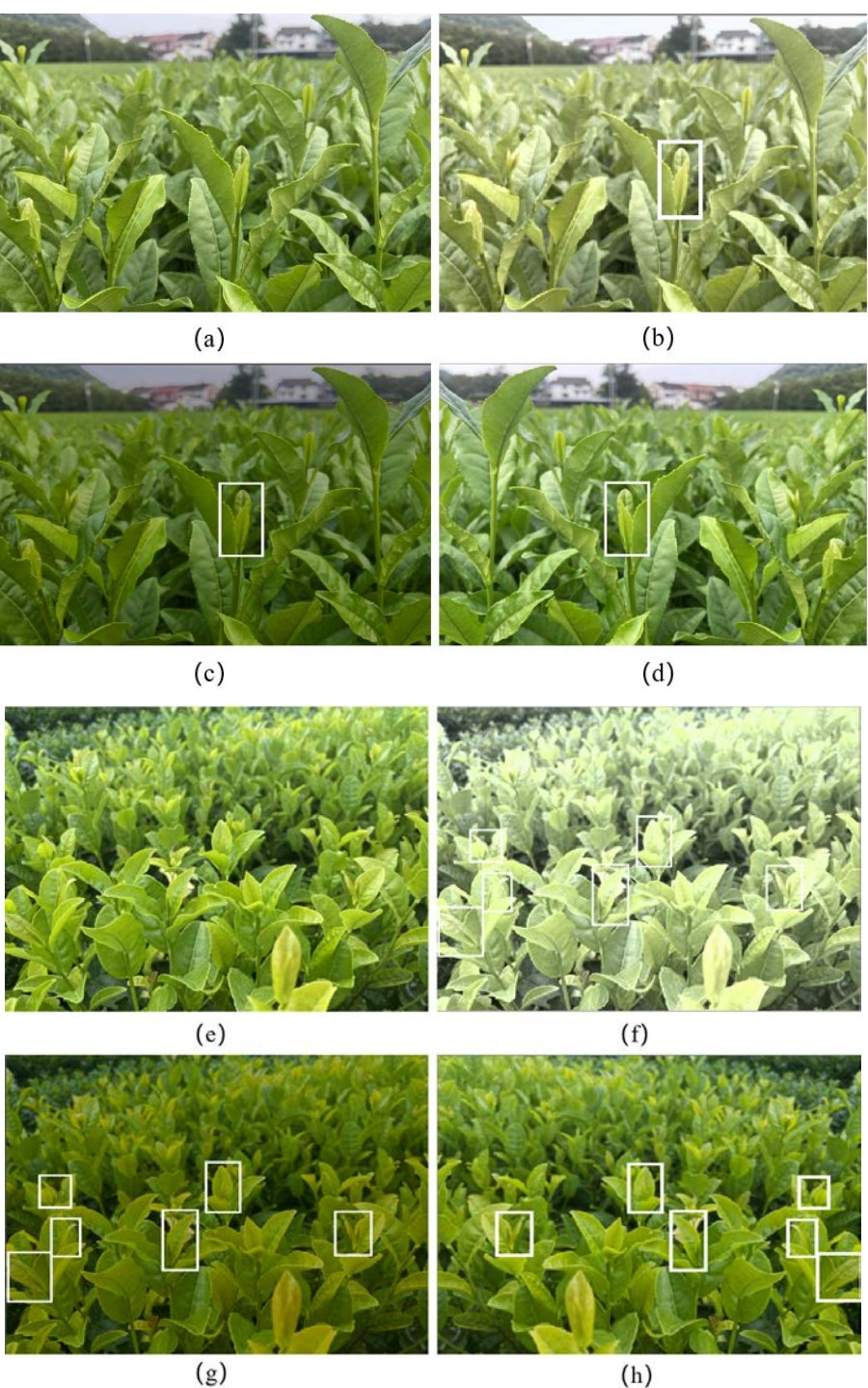

**Figure 3.** *Cont.*

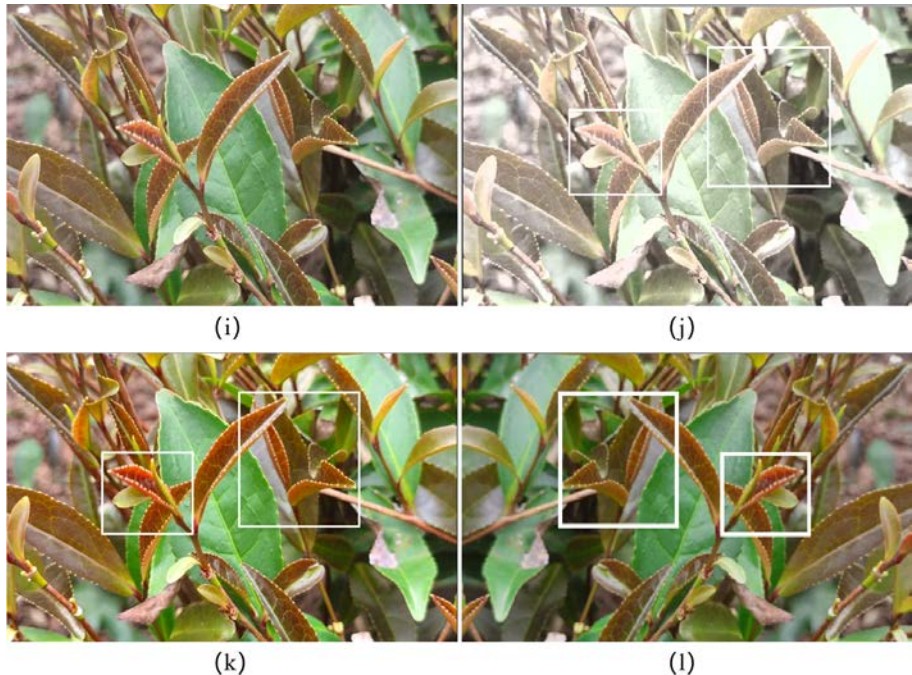

**Figure 3.** MVT dataset enhancement examples. (**a**,**e**,**i**) Image of primitive fresh tea leaves; (**b**,**f**,**j**) Image of fresh tea leaves with increased or decreased brightness; (**c**,**g**,**k**) Image of fresh tea leaves with increased or decreased color saturation; (**d**,**h**,**l**) Image of fresh tea leaves with the horizontal flip.

## 2.2. Establishment and Training of the IMVTS Model

### 2.2.1. Experimental Platform and Environmental Configuration

This study used a workstation equipped with the Ubuntu 18.04 operating system for model training. The CPU was Intel i7 12700F (Shenzhen, China) and the GPU was 3080Ti. The development environment was Pytorch 1.11. The development language was Python, and vs. code software was used for debugging.

### 2.2.2. Establishment of the IMVTS Model

This study used the YOLO v7 model as the basic framework for the recognition of multiple varieties of fresh tea leaf sprouts; the related research has shown that attention mechanisms [15] such as SENet [16], CBAM [17], and ECA [18] improved the network performance. Therefore, to improve the detection accuracy of YOLO v7's original model on the small target of fresh tea leaves, the effects of adding the CBAM or ECA attention module to YOLO v7 on the detection effect were compared; the final IMVTS network model structure obtained is shown in Figure 4. The IMVTS model mainly includes four parts: input, backbone, head, and output. First, the collected tea leaf images undergo labellmg and random data augmentation, then the tea leaf images are resized into 640 × 640 size RGB images and input to the backbone network The backbone network features the processing of fresh tea leaves with features. Second, the feature information extracted in the backbone network is combined through the characteristics of the head to obtain the characteristics of large, medium, and small sizes. Finally, the characteristics of the characteristic fusion are sent to the REP and Conv module of the head section for detection, and the final result is the output. The workflow of the algorithm is shown in Figure 5. The main network part of the YOLO v7 model is mainly composed of convolution, the extended-ELAN (E-ELAN) module, the MPConv module, and the SPPCSPC module. Among these, the E-ELAN module is based on the original ELAN, changing the calculation block while maintaining the original ELAN's transition layer structure. The network can learn more features by controlling the shortest and longest gradient path. It can resolve the relatively complicated and variable problems that can be solved for the background environment and targets appearing in the MVT dataset to extract more efficient features. The SPPCSPC module

adds parallel MaxPooling operations to a convolution series to obtain different feelings so that the algorithm can adapt to fresh tea leaf images of different resolutions and can make the network speed faster; the accuracy of detecting fresh tea leaf sprouts will also be improved. The main role of the MPConv module is to sample. The experience of the current feature layer is expanded through MaxPooling and then integrated with the normal convolutional fresh tea leaf sprout feature information. It can improve the generalization of the network, suitable for the recognition scene of multiple varieties of tea leaf sprouts [19]. When introducing the attention module, the location information and detail information of the feature information are extracted from the backbone, and there is less semantic information [20]. Feature information is easily lost after processing for small targets such as fresh tea leaf sprouts. To improve the detection accuracy of the YOLO v7 original model for fresh tea leaves, an attention mechanism was added between the backbone and the head. The structure of the attention module is shown in Figure 6. CBAM combines the channel attention mechanism with the spatial attention mechanism. The CBAM module conducts global average pooling and the largest global pooling for the input feature layer, then learns the weight information of each channel through a shared full connection layer and the Sigmoid function. Then, it learns the weight information of each point in the space through a convolutional nucleus of $3 \times 3$ and the expansion coefficient of 2, and the Sigmoid function. ECA is a channel attention mechanism. The ECA module changes the two full connection layers to one-dimensional convolution. After the Sigmoid function is passed, the channel weight information is obtained. This module avoids the dimension of information and has good cross-channel information acquisition capabilities.

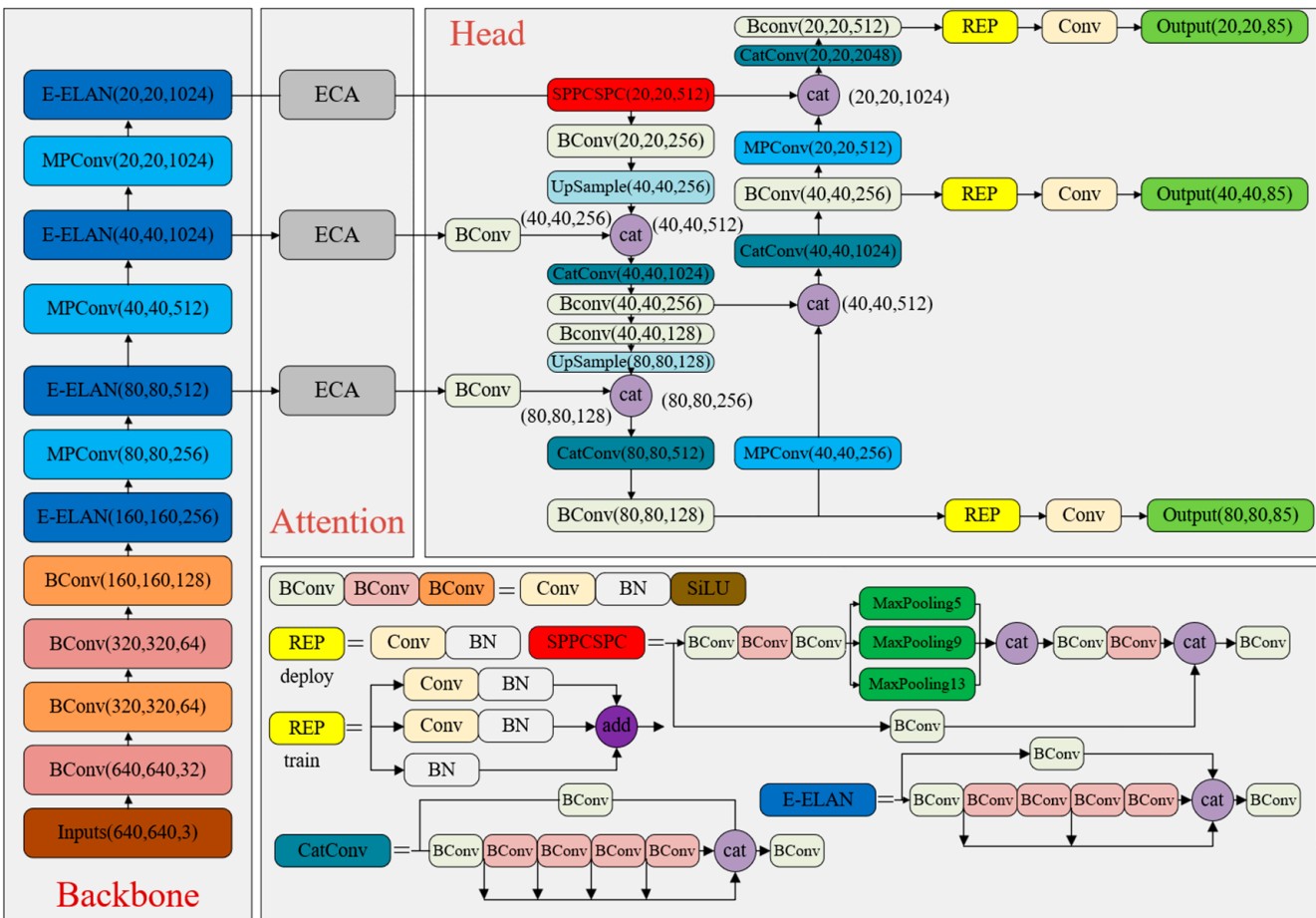

**Figure 4.** IMVTS model network structure diagram.

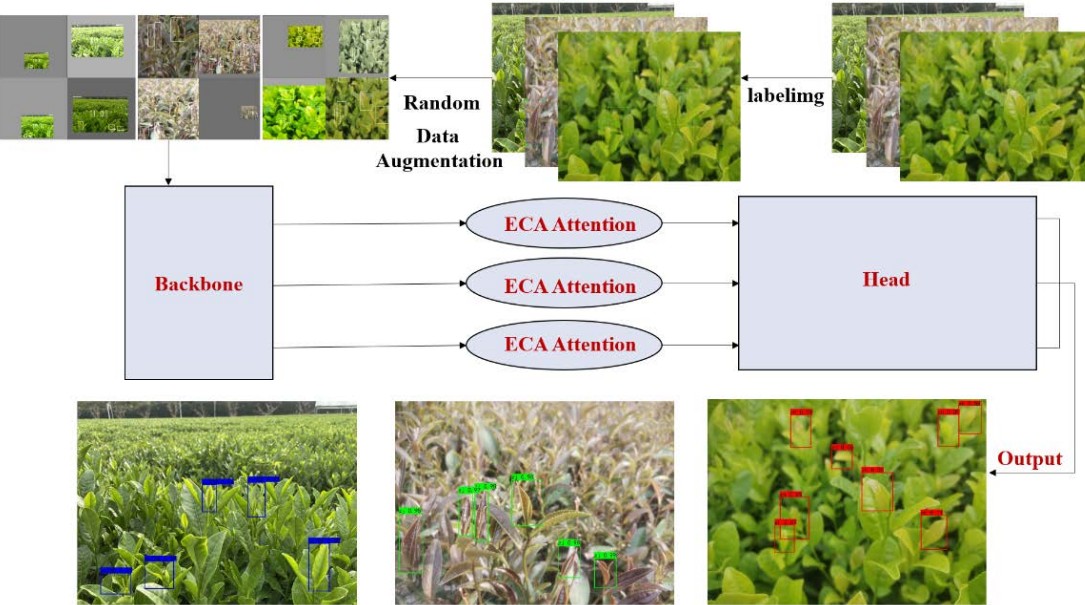

**Figure 5.** Algorithm workflow diagram.

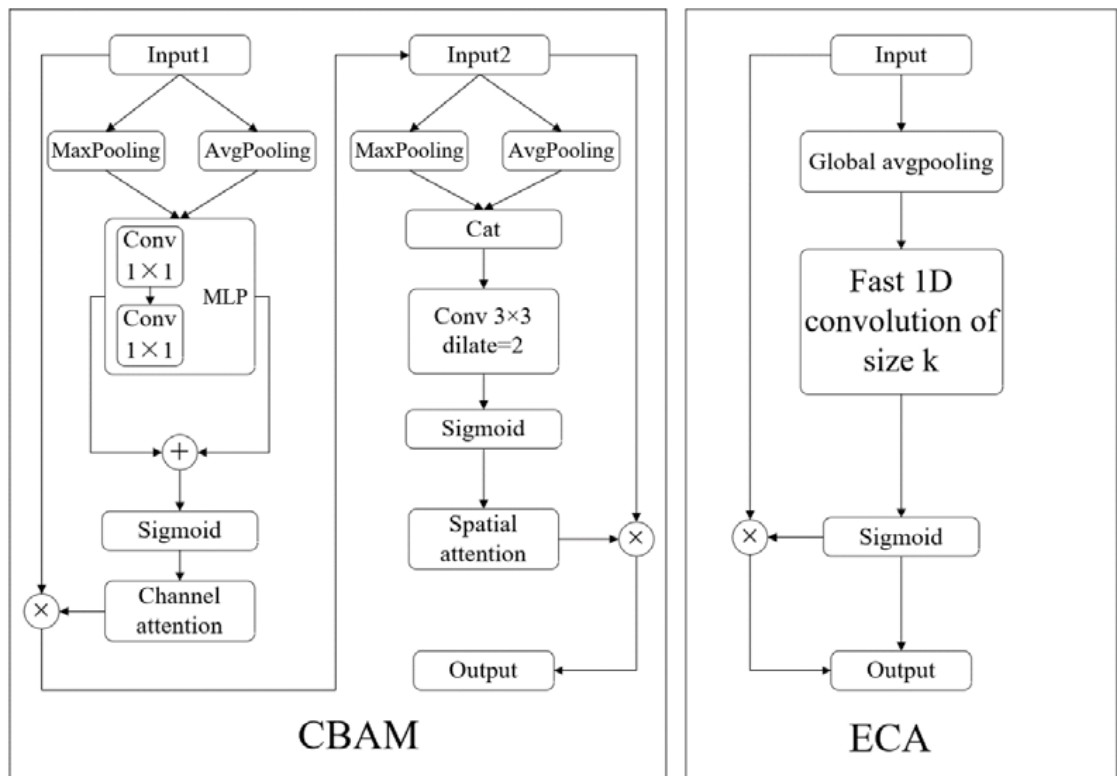

**Figure 6.** CBAM and ECA module structure diagram.

### 2.2.3. Training Parameter Settings

When training the original model of YOLO v7 and the IMVTS model, only the thawing training was performed. The thawing training set the total training generation epochs to 500; additionally, the batch size was set to 4, the initial learning rate of the model was set to 0.01, the minimum learning rate was set to 0.0001, and the SGD optimizer was used to optimize the model. The momentum parameters of the SGD optimizer were set to 0.937, and the torque annealing function was used to reduce the learning rate. In the same

environment, the training loss curves of the YOLO v7 original model, YOLO v7+CBAM model, and IMVTS model are shown in Figure 7.

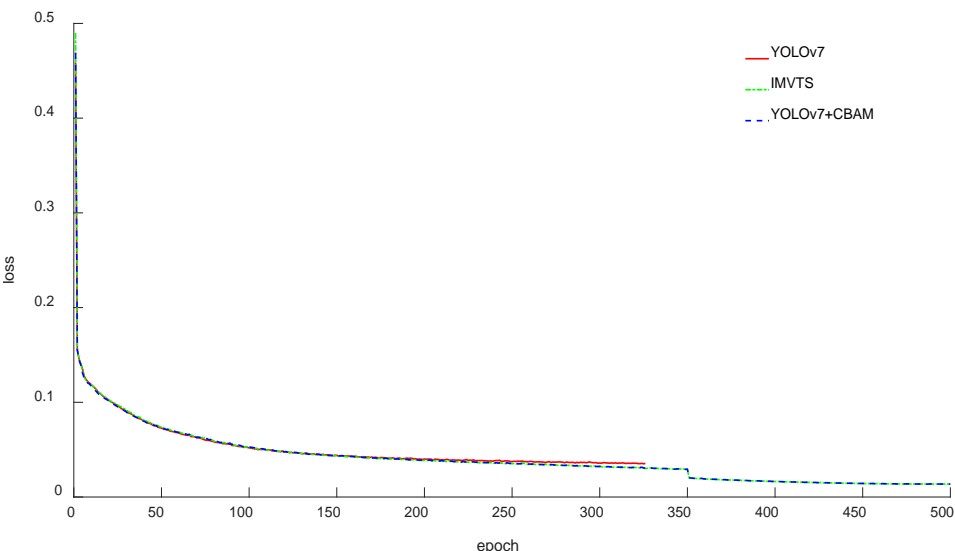

**Figure 7.** YOLO v7 original model, YOLO v7+CBAM model, and IMVTS model training loss curves.

2.2.4. Model Evaluation Index

To evaluate the detection effect of the IMVTS model, this study used precision (P), recall (R), F1 score, average precision (AP), and mean average precision (mAP) for measurement. Among these, P is the ratio of the number of correctly predicted sprouts to the total number of predicted sprouts in an image, R is the ratio of the number of correctly predicted sprouts to the total number of true sprouts in an image, the F1 score is the equalization of P and R, AP represents the accuracy of each kind of fresh tea leaf sprout recognition, and mAP represents the average value of three kinds of fresh tea leaf sprout recognition. They are calculated as follows (1)–(5):

$$P = \frac{TP}{TP + FP} \times 100\% \tag{1}$$

$$R = \frac{TP}{TP + FN} \times 100\% \tag{2}$$

$$F1 = 2\frac{P \times R}{P + R} \times 100\% \tag{3}$$

$$AP = \int_0^1 P(R)dR \times 100\% \tag{4}$$

$$mAP = \int_1^n \frac{AP_1 + AP_2 + \ldots + AP_n}{n} \tag{5}$$

where TP is the number of correctly predicted sprouts; FP is the number of falsely predicted sprouts; FN is the number of true sprouts that were not predicted as sprouts; and *n* is the number of fresh tea leaves. The values of the above parameters are obtained by counting and tallying the target detection frames in the validation sets.

## 3. Results and Discussion

### 3.1. Model Identification Results and Comparative Analysis

3.1.1. Comparison of YOLO v7 Model, YOLO v7+CBAM Model, and IMVTS Model Recognition Results

The YOLO v7 original model, YOLO v7+CBAM model, and IMVTS model were placed in the same environment for detection. The obtained model evaluation indicators are shown

in Table 1. It can be seen from Table 1 that after adding an attention mechanism module to the YOLO v7 network, the P, R, F1 score, and mAP were improved. The precision value of the IMVTS model was 99.76%, 0.10% higher than the YOLO v7+CBAM and 0.24% higher than the YOLO v7 model. The recall value of the IMVTS model was 97.03%, 0.10% higher than the YOLO v7+CBAM and 2.46% higher than the YOLO v7 model. The F1 score of the IMVTS model was 0.98, as was that of the YOLO v7+CBAM model, which was 0.01 higher than the YOLO v7 model. The mAP value of the IMVTS model was 98.82%, which was 0.02% higher than the YOLO v7+CBAM and 1.36% higher than the YOLO v7 model. In summary, the IMVTS and YOLO v7+CBAM models were better than the detection effect of the YOLO v7 model, and the IMVTS model had the best detection effect. This is because the characteristic information of fresh tea leaf sprouts is easy to lose after processing. After the ECA module is introduced, it uses a dynamic convolution nucleus to perform 1*1 convolution; it can be used to extract feature information in different areas, avoiding the number of channel dimensions caused by learning channel attention information so that the model can more effectively learn the top-level information, improving the position prediction of the fresh tea sprouts, and at the same time, can reduce the model parameters, thereby increasing the model performance. For the CBAM module, although it combines the channel attention mechanism with the space attention mechanism, there is not enough space for the information-rich feature space. For small targets, such as fresh tea leaf sprouts, the actual effect is worse than the additional ECA module.

**Table 1.** Model performance indicator assessment comparison.

| Model | P (%) | R (%) | F1-Score | mAP (%) |
|---|---|---|---|---|
| YOLO v7 | 99.52 | 94.57 | 0.97 | 97.46 |
| YOLO v7+CBAM | 99.66 | 96.93 | 0.98 | 98.80 |
| IMVTS | 99.76 | 97.03 | 0.98 | 98.82 |

3.1.2. Comparison of IMVTS Model with Mainstream Target Detection Models

To verify the advantages of the IMVTS model for multi-species tea fresh leaf shoot detection, the IMVTS model was compared with four mainstream target detection models (YOLO v3, YOLO v5, FASTER-RCNN, and SSD). The comparison experiments of the models were trained and validated under the MVT dataset. The relevant parameters were kept consistent during the experiment, and the detection effect of the model was evaluated by P, R, F1-score, and mAP, and the results are shown in Table 2. The results show that the IMVTS model proposed in this study improves in P, R, F1 values, and mAP relative to the four mainstream target detection models. Among them, IMVTS improved P by 2.38% and 1.79%, R by 7.02% and 6.31%, F1-score by 0.06 and 0.04, and mAP by 4.88% and 4.37%, respectively, relative to YOLO v3 and YOLO v5. IMVTS has higher detection accuracy than YOLO v3 and YOLO v5 because it not only inherits the advantages of the original YOLO model but also achieves higher detection speed with the same computational resources because it uses a faster convolution operation and a smaller model.

**Table 2.** Results of the model comparison test.

| Model | P (%) | R (%) | F1-Score | mAP (%) |
|---|---|---|---|---|
| IMVTS | 99.76 | 97.03 | 0.98 | 98.82 |
| YOLO v3 | 97.38 | 90.01 | 0.92 | 93.94 |
| YOLO v5 | 97.97 | 90.72 | 0.94 | 94.45 |
| FASTER-RCNN | 96.84 | 70.19 | 0.89 | 89.28 |
| SSD | 99.32 | 58.22 | 0.72 | 85.92 |

IMVTS improved P by 2.92%, R by 26.84%, F1s by 0.07, and mAP by 9.54% relative to FASTER-RCNN. The reason for the improvement in detection accuracy is that FASTER-RCNN uses resnet50 as the backbone, and its feature map only comes from the top-level

features, while the MVT dataset has small targets, and there are occlusions and blurring between targets, and only the features of the top-level of the network are used to predict the targets in a single way to extract information, which is not conducive to the localization of target frames. For IMVTS compared to SSD, P improved by 0.44%, R improved by 38.81%, F1score value improved by 0.26, and mAP improved by 12.9%. This is because the SSD model adopts a deep learning network of multi-scale characteristic fusion, but the use of low-level feature information is not enough. At the same time, the resolution of the SSD model is also low, resulting in an insufficient ability to recognize the small target SSD model, which is not conducive to conducting small target detection tasks.

3.1.3. Results and Analysis of Ablation Test

In order to verify the effectiveness of the IMVTS model proposed in this study, different optimization strategies (backbone, model size, and attention) were used in this study. An ablation test was conducted, and the comparative results were shown in Table 3. As can be seen from Table 3, the mAP of the original YOLO v7 model was 97.46%, and after the introduction of the ECA attention mechanism, the mAP increased by 1.36% and reached a peak of 98.82%. Therefore, YOLO v7 and ECA are taken as the basic framework to construct the IMVTS model proposed in this study. In addition, the ablation experiments of four mainstream target detection models (YOLO v3, YOLO v5, FASTER-RCNN, and SSD) were also conducted in this study. It can be seen from Table 3 that the IMVTS model (consisting of YOLO v7 and ECA) has the best detection effect.

**Table 3.** Results of ablation experiments.

| Model | Backbone | Model Size | Attention | mAP (%) |
|---|---|---|---|---|
| YOLO v7 | | YOLO v7 | ECA | 98.82 |
| | | | CBAM | 98.80 |
| | | | NONE | 97.46 |
| | | YOLO v7_x | ECA | 97.86 |
| | | | CBAM | 97.73 |
| | | | NONE | 96.58 |
| YOLO v3 | | | ECA | 94.26 |
| | | | CBAM | 94.20 |
| | | | NONE | 93.94 |
| YOLO v5 | Cspdarknet | YOLO v5_l | ECA | 96.03 |
| | | | CBAM | 95.66 |
| | | | NONE | 94.45 |
| | | YOLO v5_x | ECA | 95.83 |
| | | | CBAM | 95.72 |
| | | | NONE | 94.51 |
| | Convnext_tiny | YOLO v5_l | ECA | 94.28 |
| | | | CBAM | 94.23 |
| | | | NONE | 94.05 |
| | | YOLO v5_x | ECA | 94.41 |
| | | | CBAM | 94.30 |
| | | | NONE | 94.12 |
| FASTER-RCNN | Resnet50 | | ECA | 90.24 |
| | | | CBAM | 89.76 |
| | | | NONE | 89.28 |
| | Vgg | | ECA | 87.77 |
| | | | CBAM | 87.59 |
| | | | NONE | 86.33 |
| SSD | Vgg | | ECA | 88.06 |
| | | | CBAM | 87.24 |
| | | | NONE | 85.92 |
| | Mobilenetv2 | | ECA | 85.42 |
| | | | CBAM | 84.11 |
| | | | NONE | 83.31 |

### 3.1.4. IMVTS Model for VOC Dataset Detection Test

To further illustrate the effectiveness of the IMVTS model in practical engineering applications, this study uses the IMVTS model to train and test the VOC dataset, which includes images of birds, boats, buses, etc. The relevant parameters during the experiments were all kept the same as in the previous training of the MVT dataset, and the detection effect of the model was evaluated by P, R, F1-score, and mAP. Furthermore, a comparison test with YOLO v7 was conducted, and the comparison results are shown in Table 4. From Table 4, it can be seen that the IMVTS model has better detection results compared to the original YOLO v7 model.

**Table 4.** Model comparison results.

| Model | P (%) | R (%) | F1-Score | mAP (%) |
|---|---|---|---|---|
| IMVTS | 87.79 | 72.20 | 0.79 | 83.85 |
| YOLO v7 | 87.60 | 62.08 | 0.72 | 77.90 |

### 3.2. Comparison of Model Recognition Effects on the MVT Dataset

To further understand the detection effect of the models for images of fresh leaf sprouts of different varieties of tea. The YOLO v7 and the IMVTS models were used to detect the three different varieties of fresh tea leaves in the selected images. The detection results are shown in Figure 8. It can be seen that the IMVTS model detected more fresh tea leaf sprouts than the YOLO v7 model and had a better detection effect.

Figure 9 shows that when the YOLO v7 original model detects the three varieties of fresh tea leaves, there is a misunderstanding phenomenon. This is because, in the process of generating fresh tea leaf datasets, multiple fresh tea leaves and sprouts often appear in the single picture taken, and these images have problems such as blurry, chaotic positions and overlapping blocks of sprouts. As a result, the selection of the fresh tea leaf sprouts by the frame of artificial labeling is not accurate enough, resulting in error detection of the model. The IMVTS model can effectively reduce the misunderstanding phenomenon, and the specific contrast is as follows.

The results of the original model of YOLO v7, YOLO v7+CBAM, and IMVTS are given in Table 5. In Table 5, it is shown that the AP value of ZC108 can reach up to 99.87%; the AP value of ZJ can reach up to 99.64%; and the AP value of ZH is 96.97%. It can also be seen in Table 2 that compared with the ZC108 and ZJ varieties, the three models have a worse detection effect with ZH. Considering that this study used the same model for all three varieties of fresh tea leaf sprouts, the main reason should be in the images of the fresh tea leaves. Therefore, fresh leaf sprouts and old leaf images of each of the three tea varieties were selected for RGB color gamut analysis. The average and square differences of the RGB of the fresh leaf sprouts and the old leaves of the three tea varieties are shown in Figures 9 and 10. According to Figures 10 and 11, it can be seen that the differences between the three channels of the sprouts and old leaves of ZC108 and the RGB of R, G, and B are 8.33, 30.82, and 31.02, respectively; the differences between the three channels R, G, and B of the square differences of stdRGB are 2.72, 4.79, and 8.04, respectively. The differences between the three channels of the sprouts and old leaves of ZJ and the RGB of R, G, and B are 42.75, 36.52, and 33.76, respectively; the differences between the three channels R, G, and B of the square differences of stdRGB are 15.1, 13.89, and 15.35, respectively. It can be seen that there is a relatively obvious RGB gap between ZC108 and ZJ sprouts and old leaves. It is convenient for the model to distinguish the sprouts and old leaves when the model is detected. The differences between the three channels of the sprouts and old leaves of ZH and the RGB of R, G, and B are 1.34, 1.69, and 5.91, respectively; the differences between the three channels R, G, and B of the square differences of stdRGB are 2.13, 6.11, and 0.14, respectively. It can be seen that the RGB of the fresh leaf sprouts of ZH tea is very low, so the detection accuracy of the model to the fresh leaves of ZH is low.

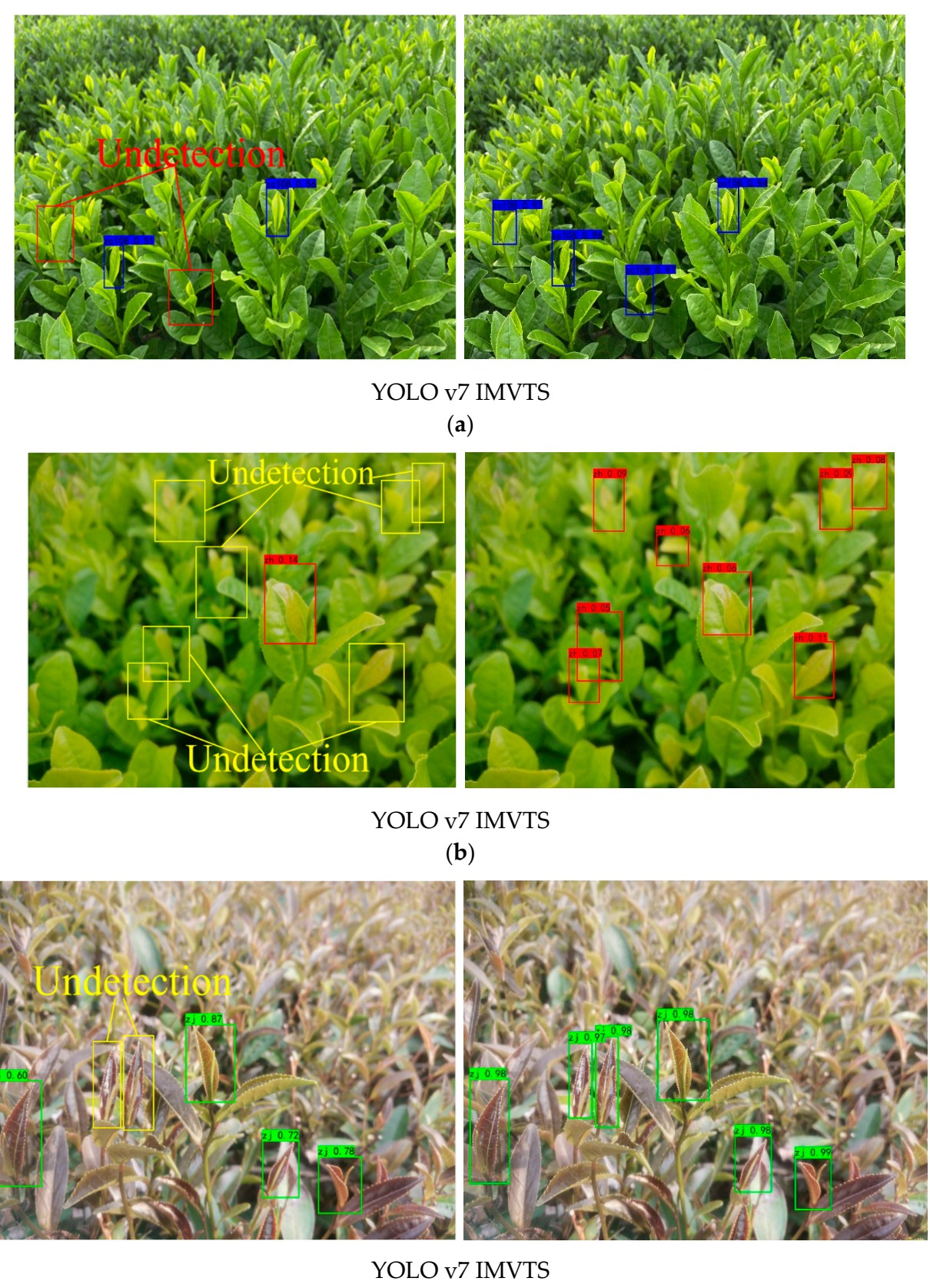

**Figure 8.** Comparison of YOLO v7 and IMVTS detection results; (**a**) ZC108; (**b**) ZH; (**c**) ZJ.

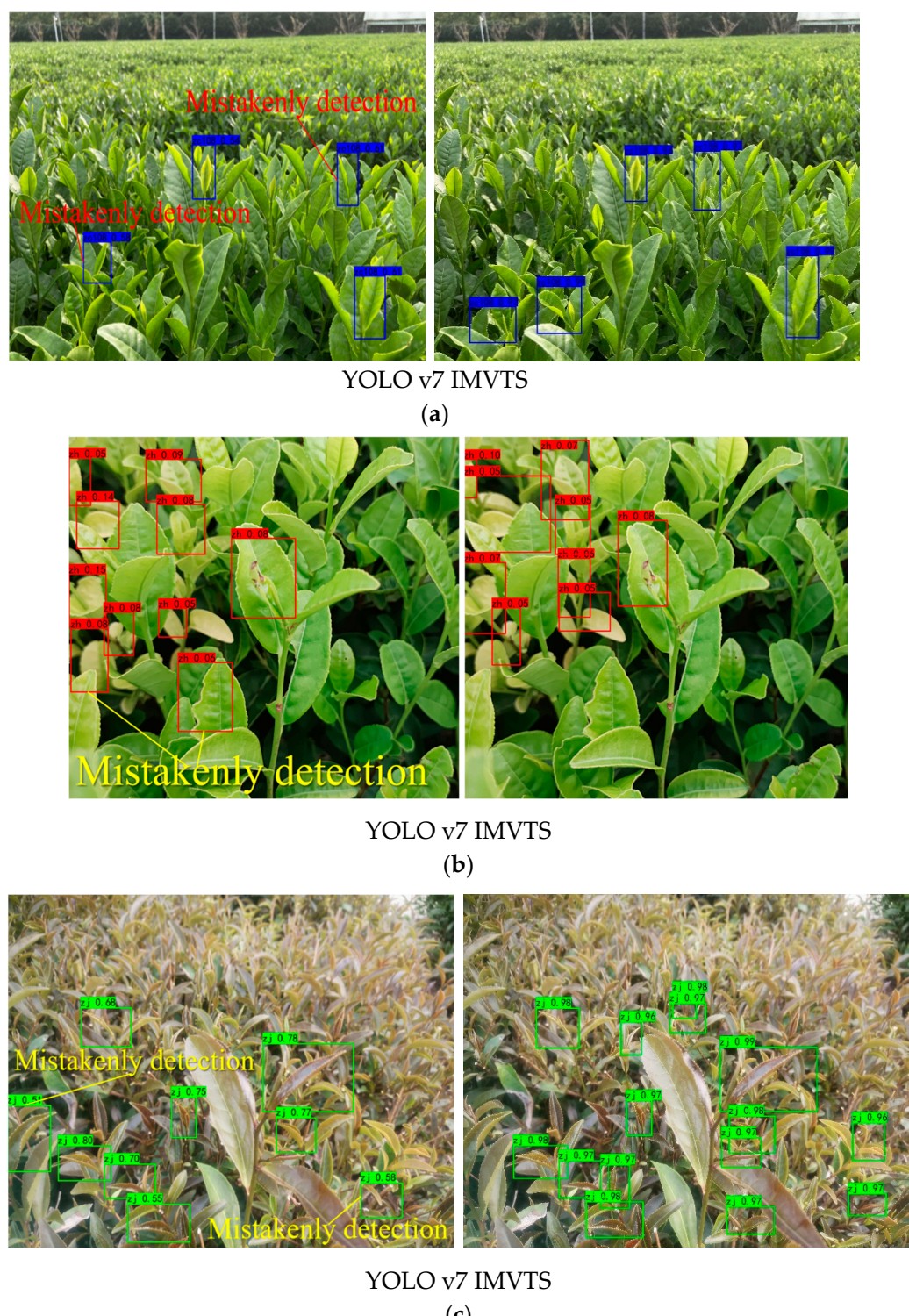

**Figure 9.** Errors in the detection of three kinds of fresh tea leaves; (**a**) ZC108; (**b**) ZH; (**c**) ZJ.

**Table 5.** Comparison of the AP results of different varieties of fresh tea leaves.

| Model | ZC108 (AP%) | ZH (AP%) | ZJ (AP%) |
|---|---|---|---|
| YOLO v7 | 99.79 | 93.30 | 99.30 |
| YOLO v7+CBAM | 99.87 | 96.89 | 99.64 |
| IMVTS | 99.87 | 96.97 | 99.64 |

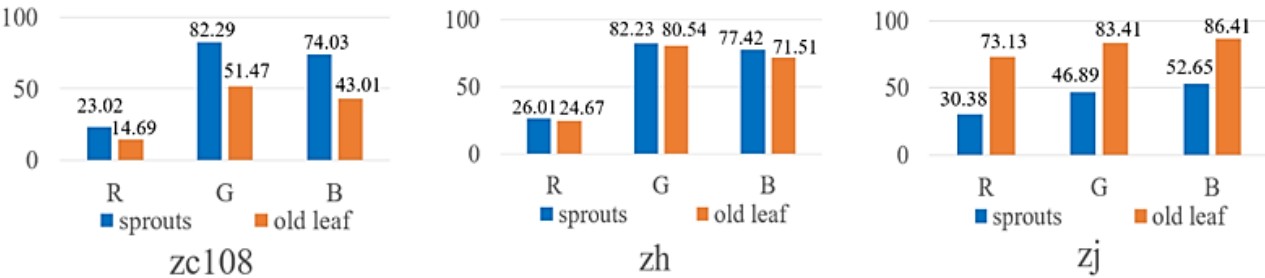

**Figure 10.** Average RGB of sprouts and old leaves of three varieties of tea.

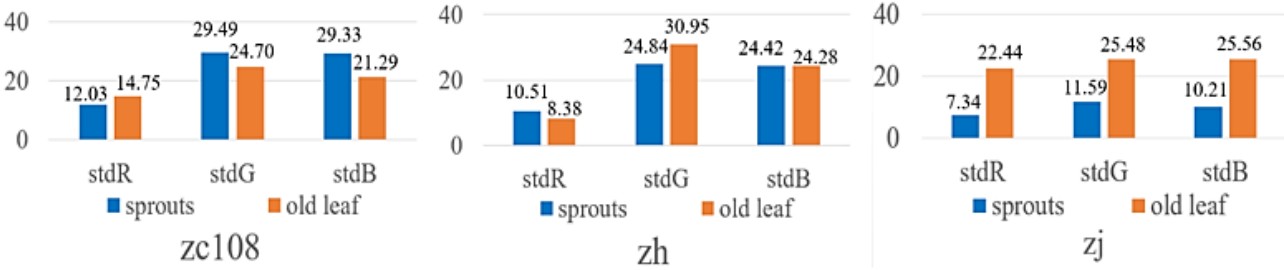

**Figure 11.** RGB variance stdRGB of sprouts and old leaves of three varieties of tea.

## 4. Conclusions

This study proposed an identification of multiple varieties of tea sprouts (IMVTS) model for the detection of fresh leaf sprouts of multiple tea species. The model training and validating dataset of multiple varieties of tea (MVT) consisted of images of three representative tea varieties (ZC108, ZH, and ZJ) in Zhejiang Province. To verify the correctness and advantages of the proposed model, IMVTS was compared with YOLO v7 and YOLO v7+CBAM. The results showed that the IMVTS model had the best detection effect, and the mean average precision (mAP) on the MVT dataset was 98.82%, which improved in comparison with YOLO v7 and YOLO v7+CBAM. In addition, this study also conducted comparison tests of the IMVTS model with mainstream target detection models (YOLO v3, YOLO v5, FASTER-RCNN, and SSD) and the IMVTS model on VOC datasets, and the results also demonstrated the superiority of the proposed IMVTS model. The average precision (AP) values of IMVTS for detecting ZC108, ZH, and ZJ tea leaves were 99.87%, 96.97%, and 99.64%, respectively. Among these, the ZH AP value was lower than those of ZJ and ZC108 because the difference in mean RGB and the difference in variance RGB between sprouts and old leaves in ZH images is smaller, and the colors of the object and background are close, making detection more difficult. In summary, the IMVTS model can improve the detection accuracy of fresh leaf sprouts of three varieties of tea, which can meet the requirements for the automatic picking of fresh leaves of the autumn famous tea and provide a basis for the future detection of fresh leaves of additional varieties of autumn tea. Future research will further focus on the improvement and design of the network structure as the IMVTS model detects objects such as ZH that are not clearly distinguished from the background.

**Author Contributions:** Writing—original draft preparation, R.Z.; methodology and software, C.L.; formal analysis, T.Y.; investigation, J.C.; data curation, Y.L.; supervision, G.L.; project administration, X.H.; writing—review and editing, Z.W. All authors have read and agreed to the published version of the manuscript.

**Funding:** This research is funded by the National Natural Science Foundation of China (Grant No. 52105284), Science and Technology Program of Meizhou, China (Grant No. 2021A0304004), Postdoctoral Science Foundation of China (Grant No. 2022M722819), and supported by China Agriculture Research System of MOF and MARA, and Key Laboratory of Crop Harvesting Equipment

Technology of Zhejiang Province (Grant No. 2021KZ04). We also thank the anonymous reviewers for their critical comments and suggestions for improving the manuscript.

**Institutional Review Board Statement:** Not applicable.

**Informed Consent Statement:** Not applicable.

**Data Availability Statement:** Data is available on request due to privacy.

**Conflicts of Interest:** The authors declare no conflict of interest.

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
