# Peer review of "IMVTS: A Detection Model for Multi-Varieties of Famous Tea Sprouts Based on Deep Learning"

_horticulturae, doi:10.3390/horticulturae9070819_

Round 1

Reviewer 1 Report

The authors explore using deep learning for tea sprouts detection in this paper. Overall, I don't see enough novelty and scientific rigor in this work.

The topic of tea sprout detection using a deep learning approach has been explored by multiple articles previously. While the authors propose a new object detection architecture IMVTS, there is no ablation study justifying the value of the proposed architecture. The improvement of 0.01 in the F1-score compared to the YOLOv7 baseline is trivial. Many reasons can result in such minor differences -- just to name a few, hyperparameter tuning, random weight initialization, and overfitting. To justify the value of IMVTS, much more comprehensive studies with widely used datasets and ablation studies need to be performed.

Applying deep learning to the tea industry in general is an interesting application domain in itself. I recommend that the authors focus on identifying and solving practical problems in the application domain itself rather than trying to come up with ad-hoc object detection architectures. 

The paper has many grammar errors and confusing expressions, which require extensive editing across the paper.

Reviewer 2 Report

In the paper IMVTS: A Detection Model for Multi-Varieties of Famous Tea Sprouts Based on Deep Learning by Runmao Zhao  et.al. the results of processing images obtained from images of tea bushes are presented. The aim of the study is an algorithm for recognizing young leaves. To my mind that research in this area is relevant, since the need for high-quality tea is great among mankind.

However, the authors should clarify some points of the study in more detail.

The description of the model in paragraph 2.2.2 “Establish an IMVTS model” contains only general phrases that do not reflect the specifics of the problem under consideration. The authors should specify what kind of data set was input and describe in detail the process of training the neural network.

Paragraph 2.2.4 introduces the parameters P and R. For their calculation, the values TP and FP are given. From the text it is completely unclear how they can be calculated. In addition, the authors should specify how these values, calculated with good accuracy, judging by the tables, are related to the real results for the determination of young leaves, which, judging by the figures, do not match reality so well.

I think that the article can be published in Horticulturae only after these comments are taken into account.

Reviewer 3 Report

The Manuscript “A detection model for multi-varieties of famous tea sprouts based on deep learning“ is an interesting article presenting the model for detection of fresh leaf sprouts from three different tea species.

Identification Multiple Varieties Tea Sprouts IMVTS (IMVTS) model was used for detection of varieties as well as differentiation between fresh and old leaves and the successfulness was compared to YOLO v7, YOLO v7+CBAM, and SSD. This model showed best detection effect and the mean Average Precision (mAP) and proved to meet the requirements of automatic picking of tea fresh leaves as well as detection of varieties of tea fresh leaves.

The manuscript is written well with methods used and results presented scientifically sound. Generally, in my opinion, the manuscript needs no revision and should be considered for publication as it is.

Round 2

Reviewer 1 Report

I appreciate the authors putting in the effort to add more baseline results for YOLOv5, Faster-RCNN, and SSD. Unfortunately, these additional experiments are very similar to the original YOLOv7 baseline experiments. They don't provide any significant value beyond the original data. There is still no ablation study justifying why IMVTS is better. I don't see meaningful deltas between this version and the previous draft. Overall, there is neither enough novelty nor scientific rigor in this work.

Reviewer 2 Report

After the revision, the authors clarified all the necessary points. I think the article should be published in Horticulturae
